# Antisense Oligonucleotide in LNA-Gapmer Design Targeting TGFBR2—A Key Single Gene Target for Safe and Effective Inhibition of TGFβ Signaling

**DOI:** 10.3390/ijms21061952

**Published:** 2020-03-12

**Authors:** Sabrina Kuespert, Rosmarie Heydn, Sebastian Peters, Eva Wirkert, Anne-Louise Meyer, Mareile Siebörger, Siw Johannesen, Ludwig Aigner, Ulrich Bogdahn, Tim-Henrik Bruun

**Affiliations:** 1Department of Neurology, University Hospital of Regensburg, 93053 Regensburg, Germany; sabrina.kuespert@ukr.de (S.K.); or; 2Department of Psychosomatic Medicine and Psychotherapy, University Medical Center Freiburg, 79104 Freiburg, Germany; 3Institute for Regenerative Medicine, Spinal Cord Injury and Tissue Regeneration Center Salzburg, Paracelsus Medical University Salzburg, 5020 Salzburg, Austria or; 4Velvio GmbH, Am Biopark 11, 93053 Regensburg, Germany

**Keywords:** TGFβ signaling, TGFBR2, antisense oligonucleotide, drug development, therapeutic frontiers

## Abstract

Antisense Oligonucleotides (ASOs) are an emerging drug class in gene modification. In our study we developed a safe, stable, and effective ASO drug candidate in locked nucleic acid (LNA)-gapmer design, targeting TGFβ receptor II (TGFBR2) mRNA. Discovery was performed as a process using state-of-the-art library development and screening. We intended to identify a drug candidate optimized for clinical development, therefore human specificity and gymnotic delivery were favored by design. A staggered process was implemented spanning *in-silico*-design, *in-vitro* transfection, and *in-vitro* gymnotic delivery of small batch syntheses. Primary *in-vitro* and *in-vivo* toxicity studies and modification of pre-lead candidates were also part of this selection process. The resulting lead compound NVP-13 unites human specificity and highest efficacy with lowest toxicity. We particularly focused at attenuation of TGFβ signaling, addressing both safety and efficacy. Hence, developing a treatment to potentially recondition numerous pathological processes mediated by elevated TGFβ signaling, we have chosen to create our data in human lung cell lines and human neuronal stem cell lines, each representative for prospective drug developments in pulmonary fibrosis and neurodegeneration. We show that TGFBR2 mRNA as a single gene target for NVP-13 responds well, and that it bears great potential to be safe and efficient in TGFβ signaling related disorders.

## 1. Introduction

Antisense oligonucleotide (ASO) technology has exceptionally developed in recent years [1]. ASOs combine essential positive drug qualities as they are at the same time highly specific and selective for their target, show long half-lives and may have rather low toxicity profiles. ASOs can be applied into anatomically defined regions, where they may be locally contained in the area of application [2,3]. Thus, Antisense technology is gaining upcoming relevance in the field of therapeutic applications [2,4,5,6,7,8]. A recent example is Nusinersen, a successful FDA-approved ASO and the first compound for treating spinal muscular atrophy [7,8,9,10,11].

In our opinion, TGFβ is a mastermind in many physiological processes and acts through the canonical SMAD signaling pathway or other non-canonical pathways [12]. However, this essential growth factor is also involved in a broad variety of pathophysiological processes, where a persistent disruption of canonical and/or non-canonical TGFβ pathways is implicated in the development of various human disorders in general, and in neurodegenerative disorders in particular.

Neurodegenerative disorders display disrupted or elevated TGFβ signaling, mainly due to chronic inflammation [13,14]. In Amyotrophic Lateral Sclerosis (ALS), TGFβ accelerates pathogenic mechanisms and has been identified as a potential key factor in pathogenesis and disease progression of ALS [15,16]. We could show that upregulated TGFβ causes stem cells to arrest in G0/G1 phase *in-vitro* and *in-vivo* [17]. Thus, stem cell regeneration is indirectly blocked by elevated TGFβ. In addition to the role of TGFβ in neurodegeneration, it is also heavily increased in patients with idiopathic pulmonary fibrosis (IPF). In IPF, aberrant production and localization of TGFβ suggests a major role in pulmonary inflammation, fibrosis, and tissue remodeling [18]. TGFβ also plays a crucial role in several tumor disorders. For instance, in metastatic pancreatic ductal adenocarcinoma, TGFβ is critical for epithelial-mesenchymal transition and thereby is responsible for disease progression and supporting metastases [19]. Currently, studies in immune-oncology show that inhibition of TGFβ might be a central target to improve therapeutic response concerning tumor immune evasion and immunotherapy [20,21].

TGFBR2 binds to TGFβ ligands, triggers TGFBR2 dimerization, and interacts with 39 different proteins. Further, TGFBR2 induces the formation of tetramers with TGFBR1-dimers and triggers phosphorylation of TGFBR1 [22,23,24]. Following activation, TGFBR1 interacts with a number of specific cytoplasmatic proteins including the SMAD proteins to influence specific gene expression. The effects of TGFBR2 and TGFBR1 signaling proverbially have “two faces” and are either beneficial (neuroprotective, stem cell proliferation, anti-inflammatory, anti-fibrotic, autophagy inducing) or detrimental (neurodestructive, stem cell arrest, pro-inflammatory, pro-fibrotic, autophagy inhibitory) depending on dose, context, and duration of activated signaling [25,26,27].

The broad range of disorders marked by upregulated TGFβ signaling highlights the need for safe and effective drugs to inhibit TGFβ signaling. Therefore, inhibition of TGFBR2 (mRNA) as the initial part of the signaling cascade was chosen as the most promising drug target for this purpose. In the current study, we identified a modified ASO with flanking locked nucleic acid wings (LNA, gapmer design) as drug candidate to specifically hybridize with the mRNA for human TGFBR2, and thus inhibit harmfully elevated TGFβ signaling.

## 2. Results

### 2.1. Discovery Process

The objective in this study was to identify a highly specific, human reactive, safe, effective, and stable antisense oligonucleotide (ASO) targeting the human TGFBR2 mRNA based on LNA gapmer technology as drug candidate for further development. We specified the ASO as a gapmer of 12 to 17 nucleotides in length, with flanking locked nucleotide acid (LNA), and a phosphorothioate (PS) backbone. In addition, the ASO should be taken up by different target cells without additional transfecting reagents (gymnotic delivery) to avoid side effects. In preparation for experimental screening rounds, an *in-silico* selection round allowed us to identify 110 ASOs of various sequences from among > 27.000 potentially effective candidates, while the first in-vitro screening round with lipofectamine-assisted delivery was performed to identify most potent candidates. In the second screening round, 30 candidates specific for human target mRNA were tested for gymnotic delivery resulting in 14 most effective candidates. Out of these, 6 candidates were further altered concerning their LNA pattern, which resulted in additional 12 LNA candidates. These collectively 18 candidates were introduced to a third screening round for highest inhibitory potency. Furthermore, we tested the most promising candidates in primary toxicity studies.

Through these extensive screenings, we identified an ASO that best fulfilled all requirements (Figure 1). This lead candidate was extensively assessed for chemical and functional stability. We designed a specific probe to assess its concentration in plasma, cerebrospinal fluid (CSF), solvents, cell pellets or tissue samples and to be used in uptake kinetics. Finally, biological function was analyzed *in-vitro* in human lung cells (adenocarcinoma epithelial cell line A549 (ATCC^®^, CCL-185 ™) and human neuronal precursor cells of cortical origin (ReNcell CX®, Millipore #SCM007).

### 2.2. In Preparation to the First Experimental Round of Screening a Total of 118 Specific and Cross-Species Reactive Antisense Oligonucleotides Against TGFBR2 Were Designed In-Silico

In a first step, a general screening procedure was performed. Among 27,693 potentially active ASO sequences with 12 to 17 nucleotides, 110 sequences were identified with regard to cross species reactivity, not binding to single nucleotide polymorphisms (SNPs), and target specificity. Besides human targets, species that are most relevant for advanced preclinical studies such as non-human primates (NHP) and rodents were also selected and introduced in the screening process (Appendix A). Three 15mer (**) and two 14mer (**) sequences were skipped because of non-acceptable predicted specificity in rodents (non-acceptable: perfect match with unintended transcript(s) / off-target(s)). In addition, X05177 (historic sequence) and two of its derivatives were not considered for selection and served as reference (Table 1).

80 candidates had superior specificity in humans and non-human primates and 30 candidates had excellent specificity in humans, non-human primates and rodents. Next, these 110 identified sequences were synthesized as LNA gapmers to further assess *in-vitro* effectivity after transfection. The human cell lines A549 and Panc-1 as well as the mouse cell line 4T1 were chosen due to their high TGFBR2 expression levels and their suitable transfection properties established in previous studies. Data were obtained by bDNA assay (branched DNA assay) in which we used cells treated with PBS and cells treated with R14082 as negative control and reference for optimal transfection and gymnotic delivery. R14082 is an ASO against Aha-4 and served as non-TGFBR2 specific ASO negative control. The positive control was directed against Aha-1 with remaining Aha-1 mRNA amounts being assessed to determine a successful bDNA assay-performance. First, A549 cells were used for transfection experiments (Lipofectamine 2000) with 20 nM of each of the 110 candidates–the level of TGFBR2 mRNA was compared to GAPDH mRNA. Using a negative control, we confirmed a specific target downregulation mediated by ASO-candidates, while our positive control confirmed general performance of the bDNA assay. Candidates producing no effect or those strongly reducing GAPDH mRNA were excluded from further analyses. Transfection of the remaining candidates at a low dose of 5 nM identified 30 effective candidates reducing TGFBR2 mRNA by over 50% (Appendix A).

### 2.3. In-Vitro Screening Identified 14 Highly Active Antisense Oligonucleotides Against TGFBR2

In a second round of screening, the remaining 30 candidates were delivered gymnotically in A549 (7.5 µM ASO), Panc-1 (5 µM ASO), and 4T1 (7.5 µM ASO) cells. Of these, 14 highly efficacious candidates–particularly in human cells – were identified and screened for potentially optimized LNA modification patterns (Figure 2). Their efficacy was defined as ability to reduce TGFBR2 mRNA compared to GAPDH gene expression and then normalized to PBS-treated cells. X05134 reduced TGFBR2 mRNA by 90% ± 0.02 after 72 h in Panc-1 cells. X05047-X05160 decreased TGFBR2 mRNA levels by 82-49% in Panc-1 cells. Less pronounced effects were observed in A549 cells for each ASO. While X05082 was more effective in 4T1 mouse cells than in the two human cells lines (mRNA reduction by 73% ± 0.02), X05135, X05084, and X05122 showed reduced efficacy in the mouse cell line compared to the human cell lines. X05160 showed no target regulation in 4T1 cells (1.05 ± 0.24) (Figure 2).

### 2.4. LNA Pattern Modifications Changed Activity of Antisense Oligonucleotides against TGFBR2

After these *in-vitro* screening rounds of sequences with standard LNA pattern of 3 LNA modifications at each wing, the 14 most active candidates were modified with different amounts of LNAs at the 5′ and the 3′ wings. For further selection the following criteria were applied: a length of 14 to 17mers, at least two base mismatches for binding to any non-target region within the human and non-human primate transcriptome, and at least one base mismatch for any non-target sequence of the transcriptome of rodents, dogs and pigs. The 6 most active parental sequences and their derivatives are listed in Table 2. Three human and non-human primate specific ASOs (X05134, X05099 and X05137), two ASOs cross-reactive against all relevant species (X05135 and X05160), and one ASO cross-reactive in human, non-human primates, and rodents (X05082) were chosen to generate derivates. Modification of parental sequence LNA pattern resulted in 12 variants with a gapmer design of 14 to 17 nucleotides in length, flanking LNA, and phosphorothioate (PTO) backbone (Table 2).

The selection of sequences from Table 2 was tested in a third round of screening for their efficacy after *in-vitro* gymnotic delivery as described above. PBS-treated cells and a negative control (R14082) served as reference and to confirm specific target downregulation after gymnotic delivery, respectively (Figure 3).

The binding sites of parental sequences and of their derivatives are displayed in Figure 3C. Results show that LNA pattern modifications changed the activity of certain sequences so that some derivatives were even more active than their parental sequence (Figure 3A): X07070, X07091 = NVP-13 and X07095 reduced TGFBR2 mRNA twice as much as their parental sequences in Panc-1 cells (X07070: 0.07 ± 0.01 vs 0.22 ± 0.01, X07091 = NVP-13: 0.12 ± 0.02 vs 0.24 ± 0.02 and X07095: 0.17 ± 0.04 vs 0.33 ± 0.04, respectively). To assess potency of the most promising ASOs, their IC50 and IC80 were determined in Panc-1 and 4T1 cells (Figure 3B). X05134, X07080 and X07091 = NVP-13 had half-maximal inhibitory concentrations below 0.3 µM (X05134: 0.23 µM, X07080: 0.18 µM and X07091: 0.26 µM respectively). In conclusion, LNA pattern modifications yielded very potent candidates in targeting TGFBR2.

### 2.5. Early In-Vivo / In-Vitro Toxicity Assessment Indicates Probable Non-Toxic Candidates

Specific and class-related toxicity was assessed in 4–6-week-old C57/Bl6N mice. A 200 µl dose of 15 mg/kg/BW was injected intravenously (i.v.) for each selected candidate at days 1, 3 and 5. Body weight and blood drawn from *vena fascicularis* were assessed at day 4 and 7 (Figure 4A). 7 days after injection alanine transaminase (ALT) and aspartate transaminase (AST) serum levels were measured (Figure 4B). In addition, remaining TGFBR2 mRNA in liver, kidney and lung were measured on day 7 (Figure 4C). Body weight did not significantly change for any ASO candidate. X05134 (ALT: 64.9 U/l ± 21.0, AST: 108.9 ± 13.5), X07080 (ALT: 287.2 U/l ± 65.3, AST: 273.4 ± 101.3) and X07095 (ALT: 95.9 U/l ± 46.7, AST: 89.1 ± 21.8, Saline ALT: 9.9 ± 2.2, Saline AST: 28.2 ± 2.2) produced relevant liver enzyme elevations while X07070, X07085, X07091 = NVP-13 and X07065 did not affect ALT and AST levels compared to saline. All mouse-reactive sequences (X07085, X07095 and X07065) resulted in highly significant target downregulation in liver (Figure 4C). For X07065, mRNA levels of TGFBR2 were also significantly reduced in lung and kidney. X07080 and X07091 = NVP-13 showed no downregulation of TGFBR2 mRNA levels in all tissues tested, which confirmed their predicted non-mouse reactivity, while upregulation of serum transaminase levels was highly significant with X07080 (Figure 4). Since X07091 = NVP-13 did not affect ALT and AST levels, it was chosen as top candidate and was further tested due to the collectively most promising effects of this ASO derivate using human primary cells *in-vitro*. Tumor necrosis factor alpha (TNFα) and interferon alpha (IFNα) secretion of human peripheral blood mononuclear cells (huPBMCs) were measured to assess unintended immunostimulatory activity after X07091 = NVP-13 treatment. After 24 h of incubation with 5 µM of X07091 = NVP-13, TNFα and IFNα levels quantified by ELISA assay were all below the limit of detection (data not shown). This qualified X07091 = NVP-13 as a safe candidate for further experimental use.

Considering effectivity, potency, and toxicity measures, X07091 = NVP-13 was chosen as lead candidate for human and non-human primate species as it was non-toxic and highly specific (Appendix A). This candidate exhibited the highest effectivity in Panc-1 cells and was the seventh most effective in A549 cells. In addition, NVP13 was the third most potent sequence, and non-toxic in huPBMCs and mice.

### 2.6. X07091 = NVP-13 Shows Time-Dependent Cellular Uptake, as well as TGFBR2 mRNA and Protein Downregulation

To assess X07091 = NVP-13 uptake by cells, we gymnotically delivered X07091 = NVP-13 to human neural progenitor cells (ReNcell CX® cells, Millipore) and A549 bronchial carcinoma cells. Treatment with 10 µM X07091 = NVP-13 showed a time-dependent uptake into cells (Figure 5A,B). Half-maximum uptake was reached after 32.77 h in A549 cells and after 15.45 h in ReNcell CX^®^ cells. On average, half-maximum concentration was reached 2.12 times faster in A549 compared to ReNcell CX^®^ cells. Maximum uptake of X07091 = NVP-13 after gymnotic delivery was about 5 times higher in A549 cells. Remarkably, the concentration of X07091 = NVP-13 in the supernatant remained stable for 72 h demonstrating a solid stability of the ASO in an *in-vitro* cell system.

The inhibitory activity of X07091 = NVP-13 on the target mRNA and following protein synthesis was also determined in ReNcell CX® cells and A549 cells. TGFBR2 mRNA and protein levels were determined at different timepoints (mRNA: 18 h, 24 h, 48 h, 72 h, 4 d, 6 d) / protein: 1 d, 4 d, 8 d, 12 d) (Figure 5C-H). mRNA levels were analyzed by quantitative real-time polymerase chain reaction (qRT-PCR) for gymnotic delivery of X07091 = NVP-13 in a concentration of 2.5 and 10 µM. In both cell lines results for both concentrations showed a stable time- and dose-dependent effect in target downregulation. At 10 µM X07091 = NVP-13, 80 % of mRNA compared to base line levels were downregulated after 1 day in A549 cells and after 2 days in ReNcell CX® cells (Figure 5C,D, data only shown for 10 µM). We therefore decided to use 10 µM X07091 = NVP-13 as effective dose for these cell lines. Further, TGFBR2 protein levels were assessed by western blot analysis (Figure 5 E–H). Our analysis showed a continuous significant intracellular TGFBR2 protein reduction from day 8 to day 12 post gymnotic delivery in A549 cells (Figure 5E,G) and ReNcell CX® cells (Figure 5F,H and Appendix A). In A549 cells, protein levels of TGFBR2 were reduced by about 50 % (0.46 ± 0.13) after day 8, and by about 70 % (0.32 ± 0.07) after day 12. Treatment of ReNcell CX® cells showed a significant reduction of the target protein after day 8 by about 35 % (0.65 ± 0.16) and then remained stable until day 12 (0.63 ± 0.17).

### 2.7. Gymnotic Transfer of NVP-13 in ReNcell CX® and A549 Cells is Efficacious even in Presence of TGFβ1

The inhibitory activity of NVP-13 in ReNcell CX® and A549 cells was also determined in presence of TGFβ1. Total incubation times were 12 d and 5 d respectively, including a pre-incubation period with TGFβ1 of 4 d (ReNcell CX®) and 2 d (A549). TGFBR2 mRNA and protein levels were determined 8 d (ReNcell CX®) and 3 d (A549) after gymnotic transfer in presence of TGFβ1. Furthermore, protein levels of pSmad2 as a marker for TGFβ signaling activity and mRNA/protein levels of Fibronectin (FN) as a functional downstream marker for a successful TGFβ signaling inhibition, were also examined (Figure 6).

NVP-13 was efficacious in downregulating TGFBR2 mRNA and protein also in presence of TGFβ1 (Figure 6A,B). In ReNcell CX® cells NVP-13 single treatment reduced the target mRNA about 86% ± 0.11 and about 82% ± 0.11 in presence of TGFβ1 in comparison to untreated cells. In A549 cells similar effects were observed (NVP-13: 84% ± 0.14, NVP-13 + TGFβ1: 93% ± 0.01). TGFBR2 reduction by NVP-13 was highly significant even in presence of TGFβ1, although high TGFβ1 level by itself led to TGFBR2 mRNA downregulation of about 41% ± 0.22 in A549 cells (Figure 6B). Downregulation of TGFBR2 protein levels also confirmed effectivity of NVP-13 for both cell lines for treatments without TGFβ1 and in presence of TGFβ1 (Figure 6). Inhibition of TGFβ signaling by gymnotic transfer of NVP-13 was verified by decreased pSmad2 protein levels in both cell lines. Furthermore, upregulated TGFβ signaling mediated by pre-incubation of TGFβ1 (ReNcell CX®: 1.42 ± 0.27, A549: 2.02 ± 0.33) was effectively inhibited by NVP-13 treatment (ReNcell CX®: 0.36 ± 0.16, A549: 1.70 ± 1.13) (Figure 6A,B). FN mRNA and protein levels were strongly upregulated by TGFβ1 treatment in ReNcell CX® (mRNA: 44.62 ± 21.50, protein: 6.36 ± 3.75) and A549 cells (mRNA: 2.82 ± 1.36, protein: 3.33 ± 2.12) and compared to those levels, were reduced by TGFβ1 + NVP-13 treatment (ReNcell CX®: mRNA: 2.79 ± 0.78, protein: 0.80 ± 0.44, A549: mRNA: 1.96 ± 1.25, protein: 2.80 ± 1.99). Inhibition of TGFβ signaling by NVP-13 was also verified by immunocytochemical staining against TGFBR2, pSmad2 and FN in ReNcell CX® and A549 cells (Figure 6C,D, Appendix A).

### 2.8. X07091 = NVP-13 Is Highly Stable in a Broad Range of Different Conditions

We tested NVP-13 samples for their stability under three different incubation conditions (−20 °C +/− 5 °C, 5 °C +/− 3 °C, 37 °C). Data shown in Figure 7A depicts a set-up of three independent experiments on purity. NVP-13 purity was measured using denaturizing Ion-Pair-Reversed-Pair High Performance Liquid Chromatography (IP-RP-HPLC) combined with UV/Mass Spectrometry (MS) or Electrospray-Ionization (ESI)/MS. In samples incubated at −20 °C for up to 12 months, no difference in NVP-13 sample concentration (content, mg/mL) and in the level of full-length product (FLP) (purity/integrity, %) was observed over time (0 month: 89.18 % ± 0.1, *n* = 2; 12 months: 89.24 % ± 0.25, *n* = 2). Thus, long-term stability of NVP-13 was confirmed for −20 °C +/− 5 °C. Incubation at 5 °C +/− 3 °C was tested for up to 1 week, in-line with prospective in-use specifications and also showed stable content and purity of NVP-13. The purity of a reference-sample of NVP-13 (*t* = 0, 0.5 mg/mL) was 89.8 %, purity for samples after 1 week were comparable with measurement variations being within the method variability (1 week: 0.57 mg/mL: 90.4 %, *n* = 1, 5.7 mg/mL: 91 %, *n* = 1). Incubation at 37 °C tested for up to 1 month also showed no significant variation across the incubation period (*t* = 0: 95.3 %, *n* = 1, 1 month: 92.4 %, *n* = 1). Measurement variations were within method variability. Conclusively, we show that content and integrity of NVP-13 in saline remained stable under different incubation conditions (Figure 7A and Appendix A). NVP-13 stability was additionally assessed after delivery into two human cell-lines. ReNcell CX^®^ and A549 cells were treated with a single dose of 10 µM NVP-13 for up to 72 h and concentration of intact NVP-13 in the respective cell culture media supernatant was measured at different time points using a specific bioprobe (Figure 6B,C). The NVP-13 concentration in cell culture media supernatant of ReNcell CX^®^ and A549 cells remained stable over 72 h (12.1 µM (1 h), 12.9 µM (72 h), and 11.7 µM (1 h), 11.5 µM (72 h), respectively). Thus, we show that NVP-13 can be considered stable for at least one year in saline at -20° C, up to 1 week at 5 °C and up to 1 month when incubated at 37 °C, and also remained intact in human cells for at least 72 h.

## 3. Discussion

The described discovery process successfully revealed a human specific and stable ASO drug candidate, NVP-13, targeting Transforming Growth Factor-β receptor II (TGFBR2) mRNA. This compound is taken up well by different cell types. NVP-13 also passed the selection process to detect early toxicity signals, which were particularly implemented to focus on candidates that will most likely prove safe in upcoming advanced studies and clinical development. This new TGFβ signaling inhibitor is chemically designed in a LNA-gapmer design with full PTO-backbone.

Most prominently, ASOs of advanced generation comprise Peptide Nucleic Acids (PNAs), N3’-P5‘phosphoroamidates (NPs) or locked nucleic acids (LNA). LNAs include a methylene bridge that connects 2’-oxygen of the ribose with 4’-carbon atoms [28,29] and are among the most promising candidates of ASO modifications. LNAs show a robust binding, nuclease resistance and a high potency in comparison to other 2’-modifications [1]. Full modification of ASOs does usually not impair the prevention of targeted mRNA translation by steric blocking but can impair or prevent degradation of targeted mRNA by RNase H [30]. LNAs induce conformational changes of DNA-RNA duplex and therefore prevent RNase H cleavage on target RNA [30]. Phosphorothioate (PS) DNAs form regular DNA-RNA duplexes, activate RNase H, carry negative charges for cell delivery and display attractive pharmacokinetic properties [31]. However, toxic PS-ASOs were described, which are able to bind directly to RNase H and serve as a competitive inhibitor. These data indicate that toxic single-stranded PS-ASOs can associate with RNase H and induce RNase H degradation [32]. Therefore, the so called ‘gapmer design‘ was established to enhance RNase H activity by a central stretch of DNA or phosphorothioate oligomers (PTO) / PS DNA monomers and flanking LNA ends, increasing stability and selectivity. Further, this design also increased target binding affinity and nuclease resistance. Therefore, we decided to modify ASO candidates in this promising LNA-gapmer design.

With prospective clinical development of a therapeutic drug in mind, the presented discovery process was implemented with focus on safety, low toxicity and efficacy at the human target sequence. Thus, in several screening rounds, ASO candidates were tested with respect to target downregulation, gymnotic uptake and primary toxicity. This approach allowed the discovery of ASOs that are highly selective and effective for target regulation with no potential side-effects of transfection agents. The latter may be of special interest for clinical development. Although the amounts needed for effective delivery with gymnotic delivery are higher compared with delivery via transfection, they are in the same range of others described before, e.g., Nusinersen, which was successfully approved by the FDA [33,34,35]. It is known that gymnotic delivery of LNA-ASOs can be highly efficient and may produce far less toxicity than standard lipofection techniques. *In-vitro* gymnotic silencing thus may allow a better prediction of *in-vivo* silencing and efficacy than lipofection [36]. Nevertheless, if toxicity would arise in further advanced studies, implementation of measures that lower the amount of a potentially toxic compound could be initiated. However, each further agent might have additional individual toxicity and potential unforeseen toxicity based on compound combination. We aim to pursue a strategy of adding complexity, i.e., potential toxicity, merely if triggered by a specific rationale.

In conclusion, after *in-silico* and *in-vitro* screening rounds and *in-vitro* / *in-vivo* primary toxicity studies for some of the most promising candidates, we identified NVP-13 as lead candidate due to highest target specificity after gymnotic delivery in combination with low toxicity profiles *in-vitro* and *in-vivo*. In addition, NVP-13 revealed high stability in several experiments testing drug degradation under various culture conditions (Figure 7).

Cell-lines that are established models for numerous indications including lung carcinoma (A549) or human neuronal precursor cells (ReNcell CX®) were used for treatment experiments to reveal the potential efficacy of NVP-13 to target several TGFβ dependent disorders. Cell uptake of NVP-13 was not identical for A549 and ReNcell CX® cells. Overall, uptake in A549 cells was significantly higher over time but half-maximal concentration was reached later in comparison to ReNcell CX® cells (Figure 5A). Our results thus also show a significant time-dependency of effective target downregulation (Figure 5C–H). Therefore, optimal concentration in target tissues will most likely depend on exposure-time and target cell type. Reduced levels of pSmad2 and FN mRNA / protein indicate functional inhibition of TGFβ signaling by NVP-13 in human neuronal progenitor cells (ReNcell CX^®^) and human lung carcinoma cells (A549) *in-vitro* (Figure 6). Based on previous findings [17], we suggest that gymnotic transfer of NVP-13 in neuronal stem cells might be an option to support adult neurogenesis and oppose neurodegeneration. Furthermore, inhibition of TGFβ signaling by NVP-13 led to reduction of fibrotic deposition (fibronectin, FN) (Figure 6). Decreasing fibronectin might also antagonize neural scarring, support anterograde- and retrograde transport of neurotransmitters and decrease fibrosis [37,38,39,40]. Reduction of pathogenic fibrotic modifications in fibrotic diseases such as pulmonary fibrosis might be another target for NVP-13 [41].

Hence, we could identify a new ASO drug candidate as TGFβ-signaling-inhibitor. Notably, the TGFβ ligand family of proteins share a remarkable set of multifunctional peptides comprising three highly homologous isoforms, TGFβ1, TGFβ2 and TGFβ3 [42]. In mammalian cells, TGFβ is produced as an inactive precursor that is subsequently processed by enzymatic cleavage. In summary, TGFβ proteins have a multitude of functions – therefore triggering this pathway is challenging for drug development. In contrast to many studies targeting ligands or TGFBR1, we intended to hit TGFβ signaling by targeting TGFBR2 mRNA as a unique novel concept [43,44].

However, many TGFβ-signaling-inhibitors are targeting TGFβ ligands or TGFBR1 [45,46]. Targeting Ligands by either monoclonal antibodies (Lerdelimumab (CAT-152/Trabio), Metelimumab (CAT-192), GC-1008 (Cambridge Antibody Technology) or ASO technology was mainly developed for treatment of fibrotic disorders and in the field of oncology (TGFβ2 antisense modified allogenic tumor cell vaccine (Lucanix, NovaRx), AP-12009 (Trabedersen) and AP-11014 (Antisense Pharma)) [43]. In addition, another approach was focused on the inhibition of TGFBR1 kinase by small-molecule inhibitors [45]. TGFBR1 specifically mediates phosphorylation of Smad2 and 3 to activate its canonical signaling pathway [47,48]. However, cross-inhibition of other core signaling pathways, low bioavailability, and severe multidrug resistance mainly led to drug failures using these compounds. To compensate for these severe side effects, encouraging further development led to polymeric scaffolds, for example [49].

With respect to the “two faces” of TGFβ, a persistent disruption of the canonical and/or non-canonical pathways is implied in the development of various human disorders in general and in neurodegenerative disorders in particular. Downregulation of TGFBR2 by NVP-13 may influence canonical and non-canonical pathways in a way so that TGFβ signaling may be balanced once again. Inhibition of TGFβ signaling by downregulating the single, specific target TGFBR2 with NVP-13 can act against multiple pathogenic processes at once. As learned from current immune-oncology concepts, such a multifactorial strategy might be essential for succeeding long lasting effects [21,50]. Based on the course of the signal cascades, we conclude TGFBR2 to represent the most vulnerable element for modulating TGFβ signaling. TGFBR1 only recognizes ligands bound to TGFBR2 but not ligands that are free in solution [50]. Proteins of the TGFβ family appear to be involved in various physiological and pathophysiological processes e.g., in the brain: increased TGFβ levels mediated by activation of TGFBR2 inhibit neurogenesis and may thus favor ageing and neurodegeneration [12,51,52]. In addition, TGFBR2 is an important TGFβ-mediator to ensure organ elasticity and plasticity by mechanical modifications of extracellular matrix and angio- or vasculogenesis, for example. [53,54]. Thus, high TGFβ levels contribute mainly to pathogenic processes [55]. Consequently, we believe that TGFBR2 represents an ideal target to act against elevated TGFβ concentrations as found in in the context of neurodegenerative disorders, such as ALS, or also in several other diseases like IPF or in immune oncology.

Currently, few approaches targeting TGFBR2 exist. A small molecule inhibitor- GW788388 is mainly a TGFBR1 inhibitor but Petersen et al. showed that also TGFBR2 was blocked by this inhibitor [56]. Pirfenidone, another small molecule, is also discussed to be involved in inhibition of TGFBR2 [57]. LY3022859, an anti-TGFBR2 IgG1 monoclonal antibody was developed by Eli Lilly Company and was investigated in several Phase I trials but was not further propagated because of clinical immune side effects (uncontrolled cytokine release) [58]. To date it remains unclear whether these side effects were due to reaction to antibody-protein epitopes or due to a complete shut-down of TGFβ signaling. In contrast, NVP-13 is no protein and only reduces the level of TGFβ signaling, thus is not susceptible for such a potential shut-down or epitope dependent immune-reaction/cytokine release.

We believe choosing mRNA as target is the most specific way for targeting TGFBR2. NVP-13 showed an effective gymnotic uptake which was a major aim of the screening process. The cellular uptake process of naked oligonucleotides (gymnotic uptake) can be divided mainly into two steps: adsorption and internalization. The mechanism of internalization is predominantly driven by endocytosis. Nonetheless, cellular uptake is a highly diverse and differentiated process that is dependent on many parameters, such as cell type, proliferation/cell cycle state, extracellular composition, oligonucleotide design/substitutions pattern, concentration, and phosphorothioate configuration [36,59,60]. Hence, no transfection agent is needed and potential additional adverse effects for patients might be avoided in clinical development, which is not always achieved [61]. Recently, we could already show promising results for application of NVP-13 in a completed 13-week GLP toxicity study in Cynomolgus monkeys (under review, [62]). Diminishing TGFβ signaling by NVP-13 showed dose-dependent upregulation of adult neurogenic niche activity. Moreover, we identified the non-canonical TGFβ-pathways to be in charge of neurogenic niche activity. As these were also dysregulated in ALS patients, we believe this to be a promising new approach for treating ALS patients (under review [62]).

In summary, NVP-13 is ready for further preclinical and clinical development, pioneering a strategy of targeting one central key pathogenic factor for many disorders. Notably, in neurologic diseases NVP-13 may allow an indirect resolution of protein aggregates by improvement of autophagy, for example. In addition, changes in extracellular matrix composition may be a prominent effect as well as normalization in immune dysregulation. Further, neurogenesis may be enhanced allowing regeneration of neurons, for example. By targeting TGFβ signaling, many pathogenic factors may be addressed at once - therefore NVP-13 has the potential to be a milestone discovery for slowing down or even functionally cure TGFβ-associated diseases and may emerge into a versatile drug in the near future.

## 4. Materials and Methods

### 4.1. TGFBR2 Antisense Oligonucleotide Design

Antisense oligonucleotides (ASOs) were designed in cooperation with Axolabs GmbH (Kulmbach, Germany). First, known TGFBR2 mRNA sequences were downloaded from the NCBI-database (version 61) and the Ensembl-database (version 73) and completed with data from the UTR database UTRdb (Nucleic Acids Res. 2010; 38 (Database issue): D75-D80.) when needed. The following sequences were used to identify potential TGFBR2 ASO sequences that are specific (no perfect hybridization with off-target sequences) and cross-species reactive for at least humans and non-human primates: human (homo sapiens: NM_003242.5, NM_001024847.2), cynomolgus monkey (macaca fascicularis: XM_005545564.1, XM_005545565.1), rhesus monkey (macaca mulatta: NM_001261151.1, ENSMMUT00000014004, ENSMMUT00000014002, ENSMMUT00000045300; utr|3MMUR075511, utr|3MMUR075513, utr|3MMUR075514, utr|5MMUR069325, utr|5MMUR069327, utr|5MMUR069329), mouse (mus musculus: NM_009371.3, NM_029575.3, ENSMUST00000035014, ENSMUST00000061101; utr|3MMUR100258, utr|5MMUR093262), rat (rattus: utr|3RNOR033157, utr|5RNOR028324), dog (canis familiaris: XM_534237.3, XM_005634331.1, ENSCAFT00000008766), and pig (sus scrofa: XM_005669322.1, XM_003132087.3, ENSSSCT00000035130, ENSSSCT00000012293). Only those sequences were selected for which there was no predicted binding to human SNPs outside the TGFBR2 mRNA (NM_003242.5) according to the NCBI dbSNP (version 138).

### 4.2. Source of TGFBR2 Antisense Oligonucleotides

ASOs were synthesized by Axolabs GmbH (Kulmbach, Germany) according to the company SOPs. DNA-phosphoramidites were purchased from SAFC Proligo (Hamburg, Germany) and LNA-phosphoramidites from Exiqon (Vedbaek, Denmark). Synthesized ASOs were purified, desalted and characterized according to the company SOPs.

For following larger quantities X07091 = NVP-13 were ordered directly at Exiqon (Vedbaek, Denmark) and manufactured at BioSpring (Frankfurt, Germany) in accordance to license conditions with Exiqon. For this batch, purity (IP-HPLC), Identity (ESI-MS) and an Endotoxin test (LAL method) were performed as part of the quality process at BioSpring (Frankfurt, Germany) (Table 3):

### 4.3. Cell Culture

For screening rounds human lung adenocarcinoma epithelial cell line A549 (ATCC^®^, CCL-185 ™), human pancreas/duct epithelioid carcinoma cell line Panc-1, (ATCC^®^ CRL-1469™), human neuronal precursor cells of cortical brain region (ReNcell CX®, Millipore Darmstadt, Germany, #SCM007) and mouse mammary gland carcinoma cell line 4T1 (ATCC^®^ CRL-2539™) were used. Cells were cultured according to manufacturer’s recommendations.

### 4.4. Discovery Process

For setting up screening conditions, A549 cells (8,000 cells/well) were transfected with 20 nM and 5 nM of each new candidate and negative control (R14082 -ASO against Aha-4 (aryl hydrocarbon receptor nuclear translocator homolog-4): Gbs Abs dTs dTs dTs dGs dTs dGs dTs dCs Abs Gbs Gb) and positive control (ASO against Aha-1: Tbs Cbs dAs dCs dAs dCs dTs dAs dAs dTs Cbs Tbs Cb) using 0.2 µl/ 96 well Lipofectamine ^®^ 2000 (ThermoFisher Scientific, Darmstadt, Germany) for 72 h without medium change. In a 2nd screening round the most active 30 candidates were used for transfection in Panc-1, A549 and 4T1 cells. To simulate a more realistic scenario for *in-vivo* studies, cells were treated without transfection agents (gymnotic conditions) in all following screenings. R14082 and PBS-treated cells were used as controls. In addition, an LNA-ASO control directed against Aha-1 served as positive control for an effective downregulation of Aha-1 to confirm successful assay-performance. Here, 10,000 cells/well of Panc-1, 8,000 cells/well of A549 and 8,000 cells/well of 4T1 (7.5 µM) cells were plated into 96-well plates. Cells were incubated with 5 µM (Panc-1) and 7.5 µM (A549, 4T1) of each ASO for 72 h without medium change before being harvested. In a 3rd screening round, 12 new derivates of 6 parental sequences of screening round 2 were tested by gymnotically delivery of A549 (7.5 µM), Panc-1 (5 µM) and 4T1 cells (7.5 µM) as described for screening round 2. For all screening rounds data were obtained after 72 h by bDNA assay.

### 4.5. Dose-Response Analysis

To determine half maximal inhibitory concentration (IC50) and minimal inhibitory concentration (IC80) best LNAs from 3rd screening round were tested in a dose-response experiment. Therefore, 8,000 cells/well of A549 cells were directly incubated with ASO candidates in a dose-range of 15, 5, 1.67, 0.56, 0.19 µM. Incubation time was 72 h without medium change. Data were obtained by bDNA assay. Calculation of IC50 and IC80 was performed by using XL fit software.

### 4.6. Timeline Analysis

X07091 = NVP-13 and a control oligo were added in concentrations of 2.5 µM and 10 µM to A549 (45,000 cells/24-well) and ReNcell CX® (55,000 cells/24-well) cells. Control oligo was designed by Biospring (Frankfurt, Germany) with the same sequence than X07091 = NVP-13 but in a scrambled sequence order (Biospring, #500175). Cells were harvested after 18 h, 1 d, 2 d, 3 d, 4 d, and 6 d after gymnotic delivery. Medium was removed and X07091 = NVP-13 and control oligo were refreshed after day 4 and 8. Cells were harvested and washed twice with PBS and subsequently used for RNA isolation. For protein analysis by western blotting X07091 = NVP-13 and control oligo were also gymnotically delivered to A549 (50,000 cells/6-well) and ReNcell CX® (75,000 cells/6-well) cells. Therefore, X07091 = NVP-13 and control oligo were added in a concentration of 10 µM to cells. Cells were harvested after 12, 8, 4, and 1 days after gymnotic delivery. Therefore, cells were washed twice with PBS and subsequently used for protein isolation.

### 4.7. Gymnotic Transfer of ReNcell CX^®^ and A549 Cells in Presence of TGFβ1

4 d (ReNcell CX^®^) or 2 d (A549) after pre-incubation in cell culture medium supplemented with TGFβ1 (ReNcell CX^®^: TGFβ1 50 ng/mL, A549: TGFβ1 10 ng/mL), medium was replaced with freshly made medium supplemented with TGFβ1. In addition, control oligo (10 μM) or X07091 = NVP-13 (10 μM) were added to initiate a gymnotic transfer. In parallel, separate samples were treated without supplementation of TGFβ1 as controls. A549 cells were harvested after 72 h. ReNcell CX^®^ cells were retreated 96 h after gymnotic transfer and incubated for another 96 h. In summary, ReNcell CX^®^ cells were harvested after 8 days exposition to control oligo (10 μM) or X07091 at the end of a 12 days incubation period. Finally, cells were washed twice with PBS and subsequently used for RNA-, protein isolation and immunocytochemistry.

### 4.8. Determination of Cellular X07091 = NVP-13 Uptake

X07091 = NVP-13 uptake kinetics were determined by measuring the concentration of X07091 = NVP-13 in cell pellets and supernatant of ReNcell CX^®^ and A549 cells at 3, 6, 12, 24, 48, and 72 h (n = 3 was pooled). Therefore, 35,000 cells per well were plated out in 24-well plates and left untreated as control or treated with 10 µM control Oligo or 10 µM X07091 = NVP-13. After harvest, the supernatant was collected, cells were counted and both, cell culture media supernatant and cell pellets, were assessed for their X07091 = NVP-13 content at Axolabs GmbH (Kulmbach, Germany) using a selective and sensitive bioprobe. X07091 = NVP-13 concentration per cell was calculated dividing cell pellet X07091 = NVP-13 concentration by the number of cells per well and uptake kinetics were calculated using the nonlinear fit function in GraphPad Prism (version7).

### 4.9. Branched DNA Assay (bDNA)

For bDNA assay cells were lysed by 150 µl/well of lysis mixture (Qantigene assay kit) diluted 1:3 with cell culture medium. mRNA expression was determined in a bDNA assay performed at Axolabs GmbH (Kulmbach, Germany) to analyze TGFBR2 mRNA reduction after ASO application. There, the Qantigene 1.0 assay kit was used for GAPDH (housekeeper) determination and the Qantigene 2.0 assay kit for TGFBR2 according to the manufacturer’s instructions. 10 µl/96 well of cell lysate were used both for GAPDH and TGFBR2 mRNA quantification.

### 4.10. Quantitative Real-Time Polymerase Chain Reaction (qRT-PCR)

Total RNA for cDNA synthesis was isolated using Innu Prep RNA Mini Kit (Analytic Jena, Jena, Germany, #845-KS-2040250) according to manufacturer’s instructions. To synthesize cDNA, total RNA content was determined using a photometer (Eppendorf, Hamburg, Germany, BioPhotometer D30 #6133000907), diluted with nuclease-free water. Afterwards first-strand cDNA was prepared with iScript cDNA Synthesis Kit (BioRad, Feldkirchen, Germany, #170-8891) and following mRNA analysis was performed as described before in Peters et al., 2017 [16]. As template 1 μl of respective cDNA was used. TGFBR1 (BioRad, qHsaCID0009475), TGFBR2 (BioRad qHsaCID0016240), TGFBR3 (BioRad, qHsaCID0009320),TGFβ1 (BioRad, qHsaCID0017026), TGFβ2 (BioRad, qHsaCID0018360), TGFβ3 (BioRad, qHsaCID0022239) and FN (BioRad, qHsaCID0012349) primer pairs were used and for relative quantification housekeeping gene GNB2L1 (BioRad, qHsaCEP0057912) was used (BioRad, Feldkirchen, Germany).

### 4.11. Western Blot

To obtain protein extracts cells were lysed using M-PER® Mammalian Protein Extraction Reagent (ThermoFisher Scientific, Darmstadt, Germany, #78501) according manufactory instructions. SDS-Acrylamid-gels (10%) were produced using TGX Stain Free^TM^ Fast Cast^TM^ Acrylamid Kit (BioRad, Feldkirchen, Germany #161-0183) according to manufacturer’s instructions. Protein samples (20 µl) were diluted 1:5 with Lämmli-buffer (6.5 µl, Roti®-Load1, Roth Karlsruhe, Germany, #K929.1), incubated at 60 °C for 30 min and loaded on the gel with the entire volume of the protein solution. Following western blot procedure was performed as described before in Peters et al., 2017 [16]. Primary Antibodies (TGFBR2: Biorbyt Cambridge, UK, Orb214665, 1:500 in ReNcell CX^®^ and 1:400 in A549 cells, pSmad2: cell signaling, cs#3104, 1:1000 for both cell lines, FN: Abcam, Cambridge, UK ab23750, 1:1000 for both cell lines) were diluted in 5% BSA. Incubation time was over night at 4°C. Second antibody (Anti-rabbit IgG, HRP-linked, 1:10,000, Cell Signaling Danvers, MA, USA, cs#12351S (1 h, RT) was added. Housekeeper-Comparison was also done with HRP-conjugated anti GAPDH (cell signaling #cs8884s, 1:2,000 in 0.5 % Blotto, 4 °C, overnight) as described before in Peters et al., 2017 [16].

### 4.12. Immunocytochemistry

ReNcell CX® and A549 cells were fixed with Roti-Histofix 4% (Roth, Karlsruhe, Germany, #P087.4) (6 min, RT), then washed three times with PBS. Cells were treated 1 h at RT with blocking solution (Zytomed, Berlin, Germany, #ZUC007-100) then incubated with respective primary antibodies listed in Table 4 and incubated at 4 °C overnight.

Subsequently, cell culture slides were washed three times with PBS following incubation with secondary antibody (1 h, RT) (Table 2). All antibody-dilutions were prepared with Antibody-Diluent (Zytomed Berlin, Germany, #ZUC025-100).

Following incubation with secondary antibody, cells were washed three times with PBS, coverslip was separated from each cell culture dish, mounted with Vectashield HardSet and treated with DAPI (Biozol, Eching, Germany #VEC-H-1500). Slides were dried overnight at 4°C, then captured by fluorescence microscopy (Zeiss, Oberkochen, Germany: Zeiss Axio Observer.Z1). Images were analyzed with Image J Software and Corel Draw X7 Software.

### 4.13. Peripheral Blood Mononuclear Cell Assay (PBMC Assay)

X07091 = NVP-13 toxicity was determined in a PBMC assay at Axolabs GmbH (Kulmbach, Germany) using TNFalpha and IFNalpha as readouts. Therefore, huPBMCS from two different voluntary donors were isolated by Ficoll gradient centrifugation (Sigma-Aldrich, Darmstadt Germany) within 40 h after blood donation and maintained in Roswell Park Memorial Institute (RPMI) 1640 Medium with L-Glutamine, 10% FBS, 5 µg/mL PHA-P and 10 ng/mL IL-3. Only blood samples tested negative for infectious agents and with normal SGPT value were used in the experiments. All cells were seeded at 100,000 cells/well into 96-well plates incubated at 37 °C and 5% CO_2_. For both readouts, huPBMCs were treated gymnotically with 5 µM of ASO for 24 h (quadruplicates). CpG Oligo (ODN2216, 0.5 µM), chol. conjugates siRNA (XD-01024, 0.5 µM) served as negative controls and poly I:C (100 µg/mL) served as positive control. Supernatants from two wells were pooled to perform an ELISA of huTNFalpha (50 µl supernatant, eBioscience Darmstadt, Germany, #BMS223INSTCE) and huIFN-alpha (20 µl, eBioscience Darmstadt, Germany, #BMS216INSTCE) in duplicates.

### 4.14. In-Vivo Toxicity Study (Mouse Model)

Early systemic toxicity of different ASOs was assessed in a mouse model at Axolabs GmbH (Kulmbach, Germany). For 7 days, body weight and alanine transaminase (ALT) as well as aspartate transaminase (AST) levels were measured. Seven promising ASOs were diluted in PBS and applied intravenously (200 µl i.v. corresponding to 15 mg/kg/BW) in four female C57/Bl6N mice of 6 weeks of age. They were treated on day 1, 3, and 5. PBS served as a negative control. On days 4 and 7, body weight was measured, and blood draw was taken from *vena fascicularis* at day 4. Mice were sacrificed on day 7 by CO_2_ narcosis. Serum from *vena cava*, liver, lung and kidney (frozen in liquid N_2_) was isolated. Tissue mRNA levels of TGFBR2 were measured in lysate of liver, kidney and lung by bDNA assay (Quantigene kit) as described above. GAPDH was used as housekeeping gene. Transaminases (ALT/AST) levels were analyzed on an COBAS Integra 400 (Roche Diagnostics, Waiblingen, Germany) from 1:10 diluted serum.

### 4.15. A bioprobe against X07091 = NVP-13

A PNA-HPLC assay for selective and sensitive bioanalysis of X07091 = NVP-13 was established at Axolabs. This assay describes an anion-exchange (AEX)-HPLC method with fluorescence detection that allows the sensitive and specific detection of the analyte oligonucleotide X07091 = NVP-13 from liquid matrices like cell lysates or plasma. The metabolite marker (X07091 = NVP-13) was provided by Exiqon (Vedbaek, Denmark) to Axolabs (Kulmbach, Germany). The assay is based on the specific hybridization of a PNA-probe that is conjugated with an Atto425-dye at the N-terminus and that is complementary to the target oligonucleotide and PNA probe with the sequence Atto425-OO- tac tgg tcc att cat g is fully complementary to parent compound.

### 4.16. Stability

Lead candidate X07091 = NVP-13 stability/integrity was assessed at BioSpring (Frankfurt, Germany) by a denaturizing Ion-Pair-Reversed-Pair High Performance Liquid Chromatography (IP-RP-HPLC) or an Ion-Pair-Reversed-Pair-Ultra-High-Performance Liquid Chromatography (IP-RP-UPLC) combined with UV/Mass Spectrometry (MS) or with Electrospray-Ionization (ESI)/MS, respectively. 5 different concentrations of X07091 = NVP-13 in sterile isotonic saline were tested. 1 mg/mL with incubation conditions at −20 °C +/− 5 °C; 0.5, 0.57 and 5.7 mg/mL at 5 °C +/− 3 °C and for 0.25 mg/mL at 37 °C. ASO content for 1 mg/mL and −20 °C +/− 5 °C was analyzed for 1, 3, 6 and 12 months; 0.5, 0.57 and 5.7 mg/mL were tested for 1 week at 5 °C +/− 3 °C and 0.25 mg/mL was tested for 1 month.

### 4.17. Statistical Analysis

GraphPad Prism (version 7) was used to calculate mean and standard deviation. * *p* ≤ 0.05, ** *p* ≤ 0.01, ± = SEM. Detailed analysis methods are listed in the respective sections.

## Figures and Tables

**Figure 1 ijms-21-01952-f001:**
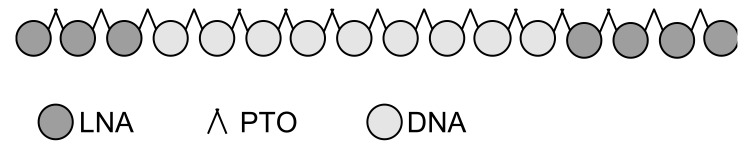
Chemical structure of the lead candidate 16mer antisense oligonucleotide. 5‘ and 3‘ wings consist of locked nucleic acids (LNA) to protect the ASO from degradation by exonucleases. LNA and deoxyribonucleic acid (DNA) are linked by a phosphorothioate backbone (PTO).

**Figure 2 ijms-21-01952-f002:**
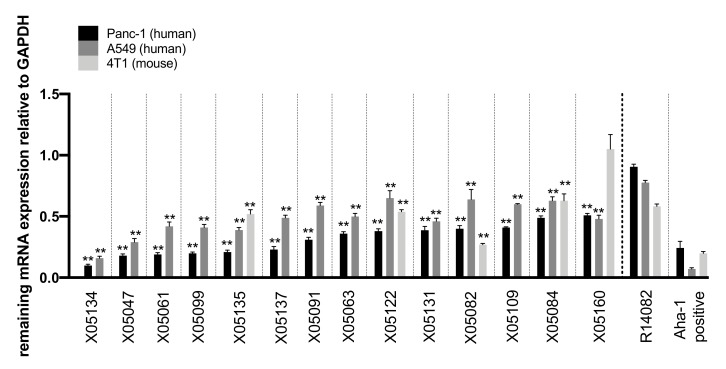
14 most active ASO candidates against TGFBR2 after gymnotic delivery. Remaining TGFBR2 mRNA after 72 h of incubation with different ASOs as measured in a bDNA assay (*n* = 4) relative to the reference gene GAPDH. TGFBR2 expression was normalized to respective PBS-treated control cells. Panc-1 cells were treated with 5 µM, A549 and 4T1 cells with 7.5 µM of each ASO. Statistics was calculated by Ordinary-one-way-Anova followed by Dunnett’s multiple comparison test. ** *p* ≤ 0.0, ± = SEM. XO = ASOs, R14082 ASO as negative control directed against Aha-4 with readout for remaining TGFBR2 mRNA. Aha-1 ASO as positive control with readout for remaining Aha-1 mRNA to determine a successful assay-performance.

**Figure 3 ijms-21-01952-f003:**
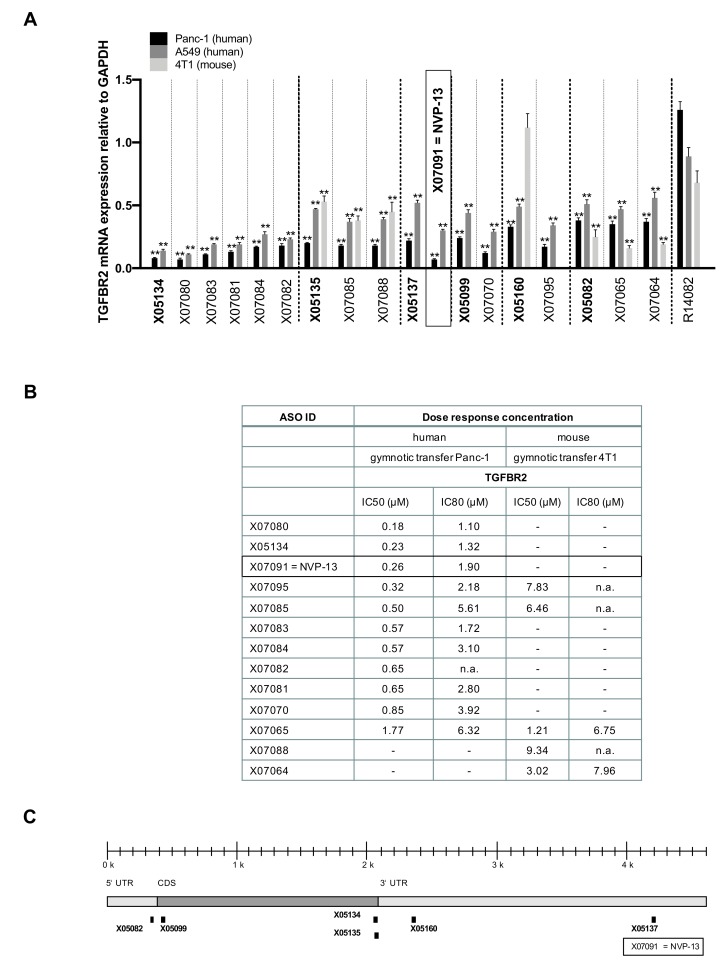
Activity, potency, and binding sites of parental sequences and their derivatives. (**A**) Activity: Remaining TGFBR2 mRNA after 72 h of incubation with different antisense oligonucleotides as measured in a bDNA assay (*n* = 4) and compared to the reference gene GAPDH and normalized to respective PBS-treated cells. Panc-1 cells were treated with 5 µM, A549 and 4T1 cells with 7.5 µM of each ASO. Some derivatives were markedly more active than their parental sequences. Statistics was calculated by Ordinary-one-way-Anova followed by Dunnett’s multiple comparison test. ** p ≤ 0.01, ± = SEM. (**B**) Potency: Inhibitory concentrations (IC50 and IC80) of the most active candidates. Panc-1 (human) and 4T1 (mouse) cells were treated gymnotically for 72 h with 5 and 7.5 µM of each candidate. Remaining TGFBR2 mRNA was assessed in a bDNA assay (*n* = 4), and IC50 and IC80 were calculated. (**C**) Human TGFBR2 mRNA binding sites of parental sequences and their derivatives.

**Figure 4 ijms-21-01952-f004:**
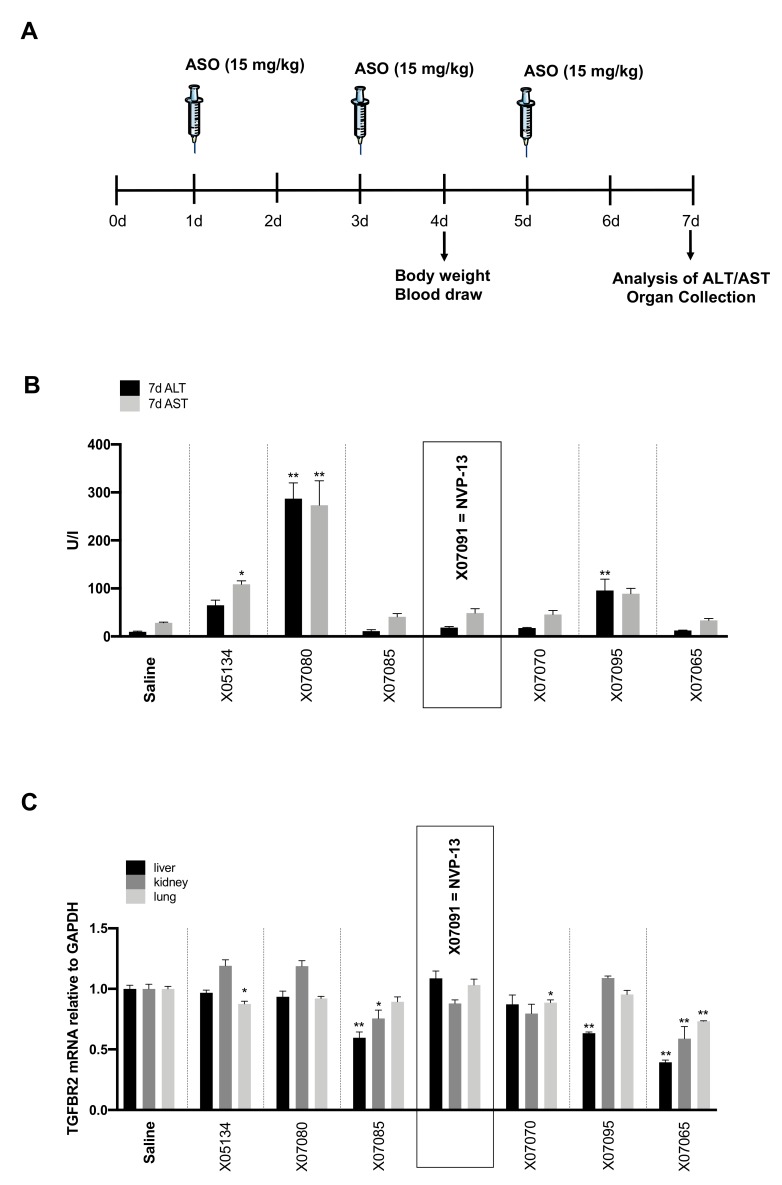
Toxicity. (**A**) Treatment procedure of *in-vivo* toxicity study. (**B**) ALT and AST levels 7 days after an intravenous (i.v.) applications of 3 × 15 mg/kg/BW (day 1, 3, 5) of each substance (*n* = 4 per candidate). Serum transaminase levels were determined by measurement on COBAS Integra 400. X05134, X07080 and X07095 showed elevated ALT/AST levels. X07085, X07091 = NVP-13, X07070 and X07065 showed no toxicity in comparison to saline. (**C**) TGFBR2 mRNA levels 7 days after i.v. application of 3 × 15 mg/kg/BW (day 1, 3, 5) of each substance (n = 4) in liver, lung and kidney measured by an bDNA assay. X07091 = NVP-13 and X07080 showed no TGFBR2 reactivity in liver, kidney and lung. Statistics was calculated by Ordinary-one-way-Anova followed by Dunnett’s multiple comparison test. * *p* ≤ 0.05, ** *p* ≤ 0.01, ± = SEM. Syringe pictogram: Servier Medical Art, CC BY 3.0 (https://creativecommons.org/licenses/by/3.0/).

**Figure 5 ijms-21-01952-f005:**
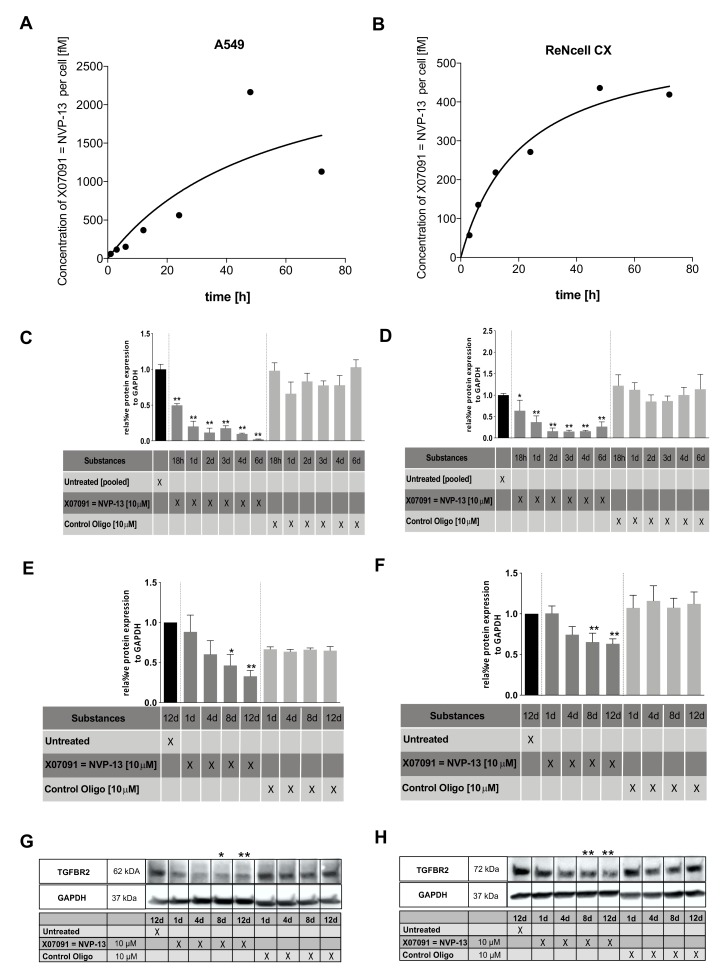
Cellular uptake, mRNA and protein kinetics of lead candidate X07091 = NVP-13. (**A** and **B**) Cellular uptake kinetics of A549 (**A**) and ReNcell CX^®^ (**B**) cells after 0–72 h of incubation with 10 µM X07091 = NVP-13 as measured with a specific bioprobe (*n* = 3). (**C** and **D**) TGFBR2 mRNA kinetics in A549 (**C**) and ReNcell CX^®^ (**D**) established in a qRT-PCR over 6 d after a single treatment with 10 µM X07091 = NVP-13 (*n* = 3). TGFBR2 mRNA levels were normalized to the housekeeper Gnb2l and normalized to untreated cells. (**E** and **F**) Densitometric analysis of TGFBR2 protein levels (Western blot) in A549 (**E**) and ReNcell CX^®^ (**F**) after 1, 4, 8, and 12 d of treatment with 10 µM X07091 = NVP-13 (n = 4 for A549 and n = 7 for ReNcell CX^®^). (**G** and **H**) Western Blot (WB) against TGFBR2 (Biorybt) in A549 (**G**) and ReNcell CX^®^ (**H**). For qRT-PCR and WB, all values were normalized to untreated controls. Statistics was calculated by Ordinary-one-way-Anova followed by Dunnett’s multiple comparison test. * *p* ≤ 0.05, ** *p* ≤ 0.01, ± = SEM.

**Figure 6 ijms-21-01952-f006:**
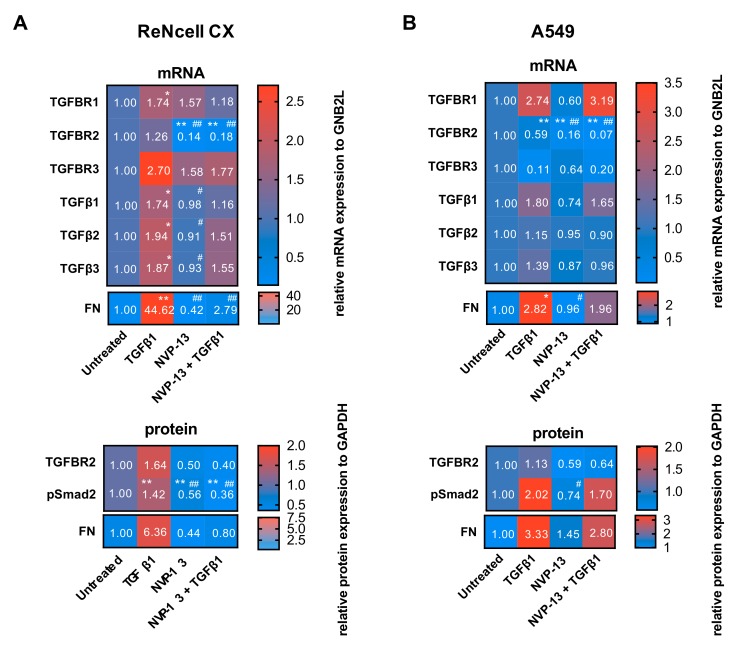
Gymnotic transfer of NVP-13 is efficacious. Heatmaps of mRNA (qRT-PCR) and protein levels (western blot) for TGFβ signaling in ReNcell CX® (**A**) and A549 cells (**B**) are shown. ReNcell CX®: mRNA and protein levels after 4 d of pre-treatment with TGFβ1 (50 ng/mL) and 8 d gymnotic transfer of NVP-13 (10 µM) in presence of TGFβ1. A549 cells were pre-treated with TGFβ1 (10 ng/mL) for 2 d followed by 3 d gymnotic transfer of NVP-13 (10 µM) in presence of TGFβ1. Quantitative RT-qPCR and Western Blotting showed a decrease of all tested proteins after gymnotic transfer of NVP-13 with TGFβ1 co-treatment. mRNA levels were normalized to the housekeeper Gnb2l and to untreated cells. Densitometric analysis of protein levels were normalized in reference to GAPDH, n ≥ 3. All statistics were calculated using the Ordinary-one-way-ANOVA followed by “Tukey’s” multiple *post hoc* comparison. (* vs. Untreated: **p* ≦ 0.05, ***p* ≦ 0.01, # vs. TGFβ1: #*p* ≦ 0.05, #*p* ≦ 0.01). Immunocytochemical staining against TGFBR2, pSmad2 and FN of ReNcell CX® cells (**C**) and A549 (**D**) confirmed observations of western blot and mRNA results. TGFBR2 (red), pSmad2 (red) and FN (green). Examination of cells was performed by fluorescence microscopy (Zeiss, Zeiss Axio^®^ Observer.Z1). Images were analyzed with Image J Software and CorelDRAW^®^ X7 Software. N = 3.

**Figure 7 ijms-21-01952-f007:**
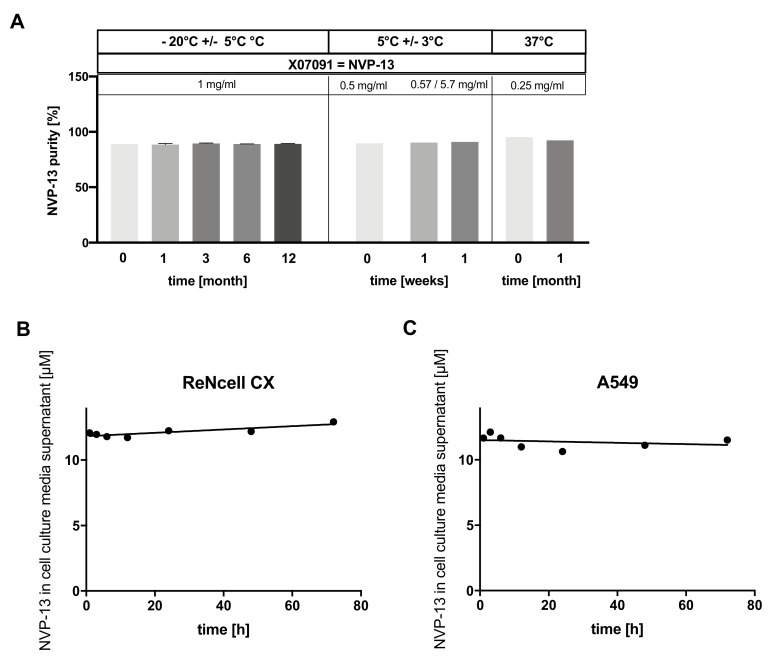
*In-use stability of lead candidate X07091 = NVP-13.* (**A**) Integrity of NVP-13 in 0.9 % NaCl (saline) when incubated with different temperature conditions and for different time intervals. For incubation at −20 °C +/− 5 °C a denaturizing Ion-Pair-Reversed-Pair High Performance Liquid Chromatography (IP-RP-HPLC) with Electrospray-Ionization (ESI)/Mass Spectrometry (MS) was used for determination of relative purity and identity of NVP-13. Aliquots incubated at 5 °C +/− 3 °C were analyzed using an IP-RP-HPLC combined with UV/Mass spectrometry and samples incubated at 37 °C (37 °C as adjusted by the integrated thermostat of New Brunswick Galaxy 170 S Incubator, Eppendorf) were analyzed using IP-RP-UPLC with UV/ESI/MS. Results showed that NVP-13 content was stable under all tested conditions. Discrepancy for content and purity/integrity were within method variability. Values are given as mean and SEM. Determination of intact NVP-13 with a specific bioprobe in the supernatant of cell culture media supernatant of ReNcell CX^®^ (**B**) and A549 cells (**C**) after incubation with 10 µM NVP-13 for up to 72 h, n = 3. Line of best fit for (**B**) and (**C**) was calculated by a nonlinear fit function.

**Table 1 ijms-21-01952-t001:** Selected candidates for 1st *in-vitro* screening round. 110 sequences with 12 to 17 nucleotides were identified as potentially active sequences against TGFBR2. ** skipped sequences because of non-acceptable specificity in rodents. X05177 = historic sequence and 2 derivates served as reference but were not considered as potential active sequences.

	Specific and X-Reactive Candidates		SUM		X05177 and Derivatives
	Human and NHP	Human, NHP and Rodent			Human, NHP and Rodent
	Number of Candidates				Number of Candidates
17mer	13	9	22	17mer	1
16mer	19	7	26	16mer	2
15mer	15	3**	18	15mer	-
14mer	7	9	16	14mer	2**
13mer	17	4	21	13mer	-
12mer	9	1	10	12mer	-
Σ	80	33	113	Σ	5

**Table 2 ijms-21-01952-t002:** LNA-Modification of parental sequences. Parental sequences with a standard modification of three flanking LNAs were modified with varying numbers of LNAs at the 5‘ and 3‘ wing yielding derivates. **X05134:** X07080-X07084, **X05135:** X07085, X07088, **X05137**: **X07091 = NVP-13**, **X05099:** X07070, **X05160:** X07095, **X05082:** X07064 and X07065. **Ab, Cb, Gb** and **Tb** = LNA, dA, dC, dG and dT = DNA, s = phosphorothioate linkage, XO = ASOs, NVP-13 = ASO lead candidate. Parental sequence IDs and LNA nucleotides are highlighted in bold.

ID	Length	Position	5′-3′ Sequence
**X05134**	16	2064	**GbsTbsAb**sdGsdTsdGsdTsdTsdTsdAsdGsdGsdGs**AbsGbsCb**
X07080	16	2064	**GbsTbsAbsGb**sdTsdGsdTsdTsdTsdAsdGsdGsdGs**AbsGbsCb**
X07083	16	2064	**GbsTbsAb**sdGsdTsdGsdTsdTsdTsdAsdGsdGsdGsdAs**GbsCb**
X07081	16	2064	**GbsTbsAb**sdGsdTsdGsdTsdTsdTsdAsdGsdGs**GbsAbsGbsCb**
X07084	16	2064	**GbsTb**sdAsdGsdTsdGsdTsdTsdTsdAsdGsdGsdGs**AbsGbsCb**
X07082	16	2064	**GbsTbsAbsGb**sdTsdGsdTsdTsdTsdAsdGsdGs**GbsAbsGbsCb**
**X05135**	16	2072	**GbsCbsTb**sdAsdTsdTsdTsdGsdGsdTsdAsdGsdTs**GbsTbsTb**
X07085	16	2072	**GbsCbsTbsAb**sdTsdTsdTsdGsdGsdTsdAsdGsdTs**GbsTbsTb**
X07088	16	2072	**GbsCbsTb**sdAsdTsdTsdTsdGsdGsdTsdAsdGsdTsdGs**TbsTb**
**X05137**	16	4217	**CbsAbsTb**sdGsdAsdAsdTsdGsdGsdAsdCsdCsdAs**GbsTbsAb**
X07091 = NVP-13	16	4217	**CbsAbsTb**sdGsdAsdAsdTsdGsdGsdAsdCsdCs**AbsGbsTbsAb**
**X05099**	15	429	**CbsGbsAb**sdTsdAsdCsdGsdCsdGsdTsdCsdCs**AbsCbsAb**
X07070	15	429	**CbsGbsAbsTb**sdAsdCsdGsdCsdGsdTsdCsdCs**AbsCbsAb**
**X05160**	17	2355	**CbsAbsGb**sGbsdCsdAsdTsdTsdAsdAsdTsdAsdAsAbs**GbsTbsGb**
X07095	17	2355	**CbsAbsGb**sdGsdCsdAsdTsdTsdAsdAsdTsdAsdAsdAs**GbsTbsGb**
**X05082**	14	355	**CbsTbsCb**sdGsdTsdCsdAsdTsdAsdGsdAs**CbsCbsGb**
X07065	14	355	**CbsTb**sdCsdGsdTsdCsdAsdTsdAsdGsdAs**CbsCbsGb**
X07064	14	355	**CbsTbsCb**sdGsdTsdCsdAsdTsdAsdGsdAsdCs**CbsGb**

**Table 3 ijms-21-01952-t003:** Purity, Identity and Quality of X07091 = NVP-13.

Molecular Formula (Sodium Salt)	C_164_H_183_O_83_N_64_S_15_P_15_Na_15_
Molecular weight	5365.3 Da (free acid) / 5693.94 Da (sodium salt)
Description	White to off-white powder, odourless
Stereochemistry	NVP-13 is a mixture of 2^15^ stereoisomers
Hygroscopicity	The drug substance is hygroscopic
Crystalline form	Amorphous
pH	The pH value of a 5 mg/mL solution of NVP-13 in - aCSF is approximately 7.1 - water for injection (WfI) is approximately 7.2, - isotonic sterile saline (0.9% NaCl) is approx. 6.5.
Solubility	The compound is a sodium salt that is soluble in aqueous solution. The solubility is ≥ 30 mg/mL

**Table 4 ijms-21-01952-t004:** Used antibodies for immunocytochemistry.

A549 cells			
Primary Antibody	Dilution	Company	Order Number
FN (rabbit)	1:50	Proteintech (St. Leon-Rot, Germany)	15613-1-AP
pSmad2 (rabbit)	1:50	Cell Signaling (Danvers, MA, USA)	cs3104s
TGF-βRII (rabbit)	1:50	Millipore (Darmstadt, Germany)	06-227
**Secondary Antibody**	**Dilution**	**Company**	**Order Number**
Cy3 goat-anti-rabbit	1:1000	Life Technologies (Darmstadt, Germany)	A10520
Alexa Fluor 488	1:1500	Life Technologies (Darmstadt, Germany)	A21441

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
