# Peer review of "Antisense Oligonucleotide in LNA-Gapmer Design Targeting TGFBR2—A Key Single Gene Target for Safe and Effective Inhibition of TGFβ Signaling"

_ijms, 2020, doi:10.3390/ijms21061952_

Round 1

Reviewer 1 Report

The manuscript describes the development of an antisense oligonucleotide (ASO) designed to down-regulate TGFR2.

There is a lot of data presented and the lead candidate does look promising bu e manuscript needs considerable work before it could be published.

The authors must engage someone to help with the English, grammar and punctuation.

Some phrases are particularly clumsy or awkward eg in the abstract "applied into restricted anatomical regions"  , the first sentence of the introduction and I am very unsure if TGFB is a mastermind.

Do all Neurodegenerative disorders display disrupted or elevated TGFB signalling (line 19)

Line 46 a LNA-ASO was developed (not discovered)

Page 3 line 32/33. Please expand on "non-acceptable predicted specificity in rodents

I am not sure if using gymnastic transfer in a couple of cell lines is going to reflect in vivo administration.

Would it not have been better to use a standard transfection agent to by-pass the delivery issue and identify the most potent compounds

Similarly, in vitro experiments cannot truly indicate safety and toxicity and I would be very cautious about making grand claims.

Page 4 line 18.  I suggest inducing that data as supplemental information.  As well as a number of sequences that were ineffective/inactive.... which takes me to another point, just because an ASO will bind to a mRNA or RNA sequence, does not mean it will exert any effect (which questions unacceptable predicted specificity in rodents earlier)

I am not convinced simply measuring absorbance will be a reliable method to assess stability.  If degraded I would have expected the base subunits to contribute to the absorbance at 260nm.  It would have been better to check these compounds out on 20% denaturing polyacrylamide gels or HPLC.

As another stability test, I would have preferred to see the ASOs incubated in human serum, 37oC for a period and then assess by PAG or HPLC.  The PS backbone and LNA wings do contribute to the stability but these compounds are not bullet-proof and will be degraded.  The DNA core would be susceptible to endonucelases.

Figure legend and labelling for Fig 6 does not seem correct

Page 13 line 8.  There is no way data in this manuscript proves the ASO is safe.  The has not been sufficient or valid toxicology undertaken.  The authors mention a GLP too study in monkeys.. why not include that data?

There have been some spectacular failures with phosphorothioate backbone ASOs.  Drisapersen was a 2OMethyl modified ASO on a PS backbone designed to correct a subclass of Duchenne mutations.  Trials were eventually halted since primary and secondary endpoints were not reached and life threatening side effects.  Some injection site reactions had not resolved after 3 years.  This same compound was well tolerated by culture human cells.  Mipomersen, another PS backbone to treat hypercholesterolemia is only prescribed in exceptional cases.  During clinical trials, at least 1/3 of participants withdrew due to side effects.  PS ASOs do not have a good safety profile.  I believe that Nusinersen, mentioned in the intro is not a fully thioated ASO.

Page 13 Line 31.  Could the authors comment on target sequence homology between human, primate, rodent, dogs, pigs

I am not sure if an 8 day 3dose study in mice would provide much information if only weight and some enzyme levels were studied.

Could the authors undertake some autophagy assays (since this was mentioned in the discussion).  Generation of in vitro stat to support function is important, rather than showing a reduction in mRNA and protein 

Author Response

Response to Reviewer 1 Comments

Point 1: The manuscript describes the development of an antisense oligonucleotide (ASO) designed to down-regulate TGFR2.

There is a lot of data presented and the lead candidate does look promising bu e manuscript needs considerable work before it could be published.

The authors must engage someone to help with the English, grammar and punctuation.

Some phrases are particularly clumsy or awkward eg in the abstract "applied into restricted anatomical regions"  , the first sentence of the introduction and I am very unsure if TGFB is a mastermind.

Response 1:

The manuscript was carefully reviewed by a native speaker colleague and revisions were integrated.

We appreciate the opinion of the reviewer concerning the term “mastermind”, though we do like it very much. We have clarified the sentence as follows: “In our opinion, TGFβ is a mastermind in many physiological processes …”

Point 2: Do all Neurodegenerative disorders display disrupted or elevated TGFB signalling (line 19)

Response 2:

Yes : “TGF-β1 is elevated acutely after injury to the brain and chronically in neurodegenerative disease.” Buckwalter MS, Wyss-Coray T. Modelling neuroinflammatory phenotypes in vivo. Journal of Neuroinflammation. 2004 Jul 1;1(1):10.

Point 3: Line 46 a LNA-ASO was developed (not discovered)

Response 3:

Changed to: “In the current study, we identified a modified ASO …”

Point 4: Page 3 line 32/33. Please expand on "non-acceptable predicted specificity in rodents

Response 4:

The definition is: “non-acceptable: perfect match with unintended transcript(s) / off-target(s)”

Point 5: I am not sure if using gymnastic transfer in a couple of cell lines is going to reflect in vivo administration.

Would it not have been better to use a standard transfection agent to by-pass the delivery issue and identify the most potent compounds

Response 5:

At page 14 we added more detail and references to that topic:

“Although the amounts needed for effective delivery with gymnotic delivery are higher compared with delivery via transfection, they are in the same range of others described before, e.g. Nusinersen, which was successfully approved by the FDA [33-35]. It is known that gymnotic delivery of LNA-ASOs can be highly efficient and may produce far less toxicity than standard lipofection techniques. In-vitro gymnotic silencing thus may allow a better prediction of in-vivo silencing and efficacy than lipofection [36]. Nevertheless, if toxicity would arise in further advanced studies implementation of measures that lower the amount of a potential toxic compound could be initiated. However, each further agent will have additional individual toxicity and potential unforeseen toxicity based on compound combination. We aim to pursue a strategy of adding complexity ,i.e. potential toxicity, merely if triggered by a specific rationale.”

Furthermore, we are not claiming that: “gymnotic transfer in a couple of cell lines is going to reflect in vivo administration”.

We designed our discovery process with the goal for gymnotic delivery because of probably less potential toxicity and with the successful clinical development of Nusinersen, a therapeutic ASO drug, in mind.

Concerning the second point, the described process had comprised “a standard transfection agent to by-pass the delivery issue and identify the most potent compounds”

See page 31 from line 23:

“First, A549 cells were used for transfection experiments (Lipofectamine 2000) with 20 nM of each of the 110 candidates (…)”

Additionally, we added more detail and clarity concerning the staggered discovery approach especially in the abstract and as outline of the results paragraph.

Point 6: Similarly, in vitro experiments cannot truly indicate safety and toxicity and I would be very cautious about making grand claims.

Response 6:

Agreed, however our claims concerning safety and toxicity are strictly confined to each step of our development paradigm concerning the in-vitro and in-vivo selection process. We don’t claim safety and toxicity for clinical applications. 

Point 7: Page 4 line 18.  I suggest inducing that data as supplemental information.  As well as a number of sequences that were ineffective/inactive.... which takes me to another point, just because an ASO will bind to a mRNA or RNA sequence, does not mean it will exert any effect (which questions unacceptable predicted specificity in rodents earlier)

Response 7:

Data was added as supplementary information (Supplementary Figure 1) as requested.

Concerning the second point: We were not claiming that ASO binding to a target will exert any effect. We have specified as one part of our rigorous selection process that if a perfect match with an unintended transcript is possible it will be preventively declared as non-acceptable.

Point 8: I am not convinced simply measuring absorbance will be a reliable method to assess stability.  If degraded I would have expected the base subunits to contribute to the absorbance at 260nm.  It would have been better to check these compounds out on 20% denaturing polyacrylamide gels or HPLC.

As another stability test, I would have preferred to see the ASOs incubated in human serum, 37oC for a period and then assess by PAG or HPLC.  The PS backbone and LNA wings do contribute to the stability but these compounds are not bullet-proof and will be degraded.  The DNA core would be susceptible to endonucelases.

Response 8:

We have revised Figure 6, the corresponding paragraph and added supplementary data (Supplementary Figure 3-5) to provide more complete and hopefully convincing data on stability. Currently we have stability data in saline and cell culture media supernatant as described in the manuscript. For later stages in development we have implemented stability experiments in CSF and Serum as well.

Point 9: Figure legend and labelling for Fig 6 does not seem correct

Response 9:

Figure 6 was revised

Point 10: Page 13 line 8.  There is no way data in this manuscript proves the ASO is safe.  The has not been sufficient or valid toxicology undertaken.  The authors mention a GLP too study in monkeys.. why not include that data?

Response 10:

We clarified the statement on safety:

“The described discovery process successfully revealed a human specific and stable ASO drug candidate, NVP-13, targeting Transforming Growth Factor-b receptor II (TGFBR2) mRNA, which is taken up well by different cell types. NVP-13 also passed the selection processes to detect early toxicity signals, which were particularly implemented to focus on candidates that will most likely prove safe in upcoming advanced studies and clinical development.”

Our claims concerning safety and toxicity are strictly confined to the stages of the in-vitro and in-vivo selection process. We don’t claim safety and toxicity for clinical applications.

The GLP study in cynomolgus monkey is part of another publication focused on in-vivo toxicity data. The corresponding section is now cited with a paper that is currently under review.

Point 11: There have been some spectacular failures with phosphorothioate backbone ASOs.  Drisapersen was a 2OMethyl modified ASO on a PS backbone designed to correct a subclass of Duchenne mutations.  Trials were eventually halted since primary and secondary endpoints were not reached and life threatening side effects.  Some injection site reactions had not resolved after 3 years.  This same compound was well tolerated by culture human cells.  Mipomersen, another PS backbone to treat hypercholesterolemia is only prescribed in exceptional cases.  During clinical trials, at least 1/3 of participants withdrew due to side effects.  PS ASOs do not have a good safety profile.  I believe that Nusinersen, mentioned in the intro is not a fully thioated ASO.

Response 11:

We do appreciate this comment and agree that there have been failures with ASO in the past. Particularly with respect to the history of this drug class, we had designed our screening process accordingly.

Concerning the chemistry of Nusinersen please see:  “Nusinersen, FDA approved in 2016, is a 2′-O-MOE PS SSO,(…). (PS = phosphorothioate).

(Smith CIE, Zain R. Therapeutic Oligonucleotides: State of the Art. Annu Rev Pharmacol Toxicol. 2019 Jan 6;59:605–30.)

Point 12: Page 13 Line 31.  Could the authors comment on target sequence homology between human, primate, rodent, dogs, pigs

Response 12:

No, we did not evaluate target homologies or alignments between these species.

Point 13: I am not sure if an 8 day 3dose study in mice would provide much information if only weight and some enzyme levels were studied.

Response 13:

This study was not supposed to be a complete toxicity study. Primary experiments on toxicity were one piece of a strategy to detect toxicity signals early-on in the discovery process. This was preventively implemented as standard procedure in terms of a diligent ASO discovery process.

Point 14: Could the authors undertake some autophagy assays (since this was mentioned in the discussion).  Generation of in vitro stat to support function is important, rather than showing a reduction in mRNA and protein 

Response 14:

Functional data downstream of TGF-beta signalling will be part of another publication with focus on these topics in particular.  

Reviewer 2 Report

The paper reports development of the specific PS-LNA-gapmer antisense oligonucleotide directed towards human mRNA of the transforming growth factor receptor 2 (TGFRII) target, as a new lead compound for silencing of gene whose expression is elevated in various human disorders. The process of selection was based on the three screening steps. The first screening round was done in silico and allowed to identify 110 ASOs of various sequences (12-12 nt long) from among >27K potentially active sites, while in vitro screening with lipofectamin-assisted delivery performed in two human and one mouse cell lines, allowed to identify 30 candidates, specific for human target mRNA (active at rather low concentration of 5 nM). In the second screening round the authors proved that gymnotic delivery of this phosphorothioate - LNA ASO is possible for 14 candidates, while optimized LNA modification pattern delivered 12 variants of gapmers. The third screening round delivered further optimized LNA candidates with the highest inhibitory potency, non-toxic in mouse model. The lead NVP-13 compound was selected for further testing in human primary cells in vitro. Time-dependent cellular uptake of ASO has shown in human neuronal progenitor cells and in A549 bronchial carcinoma, although the latter cells easier accepted ASO than primary cells. The authors present efficient silencing activity of the selected ASO and its high stability at various temperatures. The methodology is well described, and the references are used properly.

I have some suggestions / questions before the paper can be accepted for publication:

Q1: It would be good to let the readers know what was the idea behind for in silico screening of ASOs for wide range of species, other than human and eventually animal models related to advanced preclinical studies (mouse, monkey?).

Q2: Neither in the Results nor in the Discussion section the authors comment the 3 orders of magnitude higher concentration of ASO needed for gymnotic delivery comparing delivery through transfection. Is this observation advantageous for clinical applications?

Q3: Why polyanionic ASOs enter the cell without any transfection assistance? It would be good to discuss the mechanism of gymnotic uptake of the PS-ASOs, and give some references.

Q4: The kind of solvent in which stability of ASO is checked is missing in the text. Also, It is not clear why stability of ASO in 37oC is higher than in 5oC. Generally, this stability section is not clearly written, and it is difficult to follow the cited results. Do you mean “cell medium” when writing “Supernatant”?

Q5: p.13, l.18: please correct the merit of the sentence “Modification of flanking…” – “degradation of mRNA via steric blocking” is not properly defined, steric blocking impairs translation, but does not cause the mRNA degradation.

Q6: no discussion is given how LNA and PS-modifications influence the RNase H activity. PS-backbone impairs RNase H activity!

Q7: The reference is missing to the recent data on NVP-13 study in monkey (Discussion chapter).

Q8: Discussion chapter is too long. Please consider shortening the text.

Minor concerns:

p.2, l. 6 Remove “extremely”

p.2, l.23 “We could show..” ref 20? Correct the list of authors.

p.4, l.13/14 Reword sentence starting with “TGFRII…” for “The level of TGFRII mRNA was compared …”

12, l. 8 Remove “stored at”

p.23, Fig. 6 – correct Celsius centigrade abbreviation – it should be °C, and not 0C

Author Response

Response to Reviewer 2 Comments

The paper reports development of the specific PS-LNA-gapmer antisense oligonucleotide directed towards human mRNA of the transforming growth factor receptor 2 (TGFRII) target, as a new lead compound for silencing of gene whose expression is elevated in various human disorders. The process of selection was based on the three screening steps. The first screening round was done in silico and allowed to identify 110 ASOs of various sequences (12-12 nt long) from among >27K potentially active sites, while in vitro screening with lipofectamin-assisted delivery performed in two human and one mouse cell lines, allowed to identify 30 candidates, specific for human target mRNA (active at rather low concentration of 5 nM). In the second screening round the authors proved that gymnotic delivery of this phosphorothioate - LNA ASO is possible for 14 candidates, while optimized LNA modification pattern delivered 12 variants of gapmers. The third screening round delivered further optimized LNA candidates with the highest inhibitory potency, non-toxic in mouse model. The lead NVP-13 compound was selected for further testing in human primary cells in vitro. Time-dependent cellular uptake of ASO has shown in human neuronal progenitor cells and in A549 bronchial carcinoma, although the latter cells easier accepted ASO than primary cells. The authors present efficient silencing activity of the selected ASO and its high stability at various temperatures. The methodology is well described, and the references are used properly.

I have some suggestions / questions before the paper can be accepted for publication:

Point 1: Q1: It would be good to let the readers know what was the idea behind for in silico screening of ASOs for wide range of species, other than human and eventually animal models related to advanced preclinical studies (mouse, monkey?).

Response 1:

We added information beginning at page 4 line 2:

“Besides human targets, species that are most relevant for advanced preclinical studies such as non-human primates (NHP) and rodents were also selected and introduced in the screening process”

Point 2: Q2: Neither in the Results nor in the Discussion section the authors comment the 3 orders of magnitude higher concentration of ASO needed for gymnotic delivery comparing delivery through transfection. Is this observation advantageous for clinical applications?

Response 2:

We added the following section at page 14 line 1:

“Although the amounts needed for effective delivery with gymnotic delivery are higher compared with  delivery via transfection, they are in the same range of others described before, e.g. Nusinersen, which was successfully approved by the FDA [33-35]. It is known that gymnotic delivery of LNA-ASOs can be highly efficient and may produce far less toxicity than standard lipofection techniques. In-vitro gymnotic silencing thus may allow a better prediction of in-vivo silencing and efficacy than lipofection [36]. Nevertheless, if toxicity would arise in further advanced studies implementation of measures that lower the amount of a potential toxic compound could be initiated. However, each further agent will have additional individual toxicity and potential unforeseen toxicity based on compound combination. We aim to pursue a strategy of adding complexity ,i.e. potential toxicity, merely if triggered by a specific rationale.    “       

Point 3: Q3: Why polyanionic ASOs enter the cell without any transfection assistance? It would be good to discuss the mechanism of gymnotic uptake of the PS-ASOs, and give some references.

Response 3:

We added the following section at page 16 line 10:

“The cellular uptake process of naked oligonucleotides (gymnotic uptake) can be divided mainly into two steps: adsorption and internalization. The mechanism of internalization is predominantly driven by endocytosis. Nonetheless, cellular uptake is a highly diverse and differentiated process that is dependent on many parameters, such as cell type, proliferation/cell cycle state, extracellular composition, oligonucleotide design/substitutions pattern, concentration, and phosphorothioate configuration [36,54,55].”

Point 4: Q4: The kind of solvent in which stability of ASO is checked is missing in the text. Also, It is not clear why stability of ASO in 37oC is higher than in 5oC. Generally, this stability section is not clearly written, and it is difficult to follow the cited results. Do you mean “cell medium” when writing “Supernatant”?

Response 4:

The related section was revised and written more clearly. We have also revised Figure 6 and added supplementary data (Supplementary Figure 3-5) to provide more complete and hopefully convincing data on stability.

“Supernatant” was more clearly written as “supernatant of cell culture media” or “cell culture media supernatant”.

Point 5: Q5: p.13, l.18: please correct the merit of the sentence “Modification of flanking…” – “degradation of mRNA via steric blocking” is not properly defined, steric blocking impairs translation, but does not cause the mRNA degradation.

Response 5:

The sentence was changed into:

“Full modification of ASOs does usually not impair the prevention of targeted mRNA translation by steric blocking but can impair or prevent degradation of targeted mRNA by RNase H [30].”

Point 6: Q6: no discussion is given how LNA and PS-modifications influence the RNase H activity. PS-backbone impairs RNase H activity!

Response 6:

At page 14 line 3 the following paragraph was added concerning the influences on RNase H:

“[30]. LNAs induce conformational changes of DNA-RNA duplex and therefore prevent RNase H cleavage on target RNA [30]. Phosphorothioate (PS) DNAs form regular DNA-RNA duplexes, activate RNase H, carry negative charges for cell delivery and display attractive pharmacokinetic properties [31]. However, toxic PS-ASOs were described, which are able to bind directly to RNase H and serve as a competitive inhibitor. These data indicate that toxic single-stranded PS-ASOs can associate with RNase H and induce RNase H degradation [32].”

Point 7: Q7: The reference is missing to the recent data on NVP-13 study in monkey (Discussion chapter).

Response 7:

A reference was added.

Point 8: Q8: Discussion chapter is too long. Please consider shortening the text.

Response 8:

The discussion chapter was revised.

Point 9: Minor concerns:

p.2, l. 6 Remove “extremely”

p.2, l.23 “We could show..” ref 20? Correct the list of authors.

p.4, l.13/14 Reword sentence starting with “TGFRII…” for “The level of TGFRII mRNA was compared …”

12, l. 8 Remove “stored at”

p.23, Fig. 6 – correct Celsius centigrade abbreviation – it should be °C, and not 0C

Response 9:

All minor concerns were edited.